# Seed Source for Restoration: Little Bluestem (*Schizachyrium scoparium* (Michx.) Nash) and the Carolina Sandhills

Elizabeth Johnson *, Althea Hotaling Hagan * and Patrick Hiesl 

Department of Forestry and Environmental Conservation, Clemson University, Clemson, SC 29634, USA
* Correspondence: ellie.johnson@macd.org (E.J.); shotali@clemson.edu (A.H.H.)

**Abstract:** Prairie and savanna ecosystems have declined dramatically worldwide. In the Southeastern United States, longleaf pine savannas have been reduced to less than 3% of their pre-European range. Restoring longleaf pine across the area has become a regional goal. Little bluestem (*Schizachyrium scoparium* (Michx.) Nash) is critical to carrying the ecologically important fire through this ecosystem in some longleaf pine savannas. Little bluestem has a range that spans most of north America and is thought to display ecotypic variation. As a part of a longleaf pine restoration project in Camden, SC, we investigated whether the seed source of little bluestem and the site preparation techniques impacted the survival and growth of broadcasted seeds. In the field and greenhouse, we compared locally and commercially sourced seeds and field site preparation techniques including discing, raking, or no treatment. At the end of the growing season, there were significantly more plants grown from seeds collected locally compared to plants from seeds available commercially. Plants grown from locally collected seeds also invested more heavily in roots than plants grown from commercial seeds. Site preparation techniques did not appear to significantly impact plant survival. Collecting seed locally will help to ensure long-term restoration success by establishing populations of plants that are adapted to the local environmental conditions.

**Keywords:** little bluestem; longleaf pine savannah; ecotype; restoration; seed source

## 1. Introduction

Prairies and savannas were once a dominant ecosystem type across the globe, composed of a wide diversity of highly productive bunchgrasses, legumes, and composites [1–3]. They traversed over a myriad of environments with differing topographical, hydrological, and physiochemical conditions [1,4,5]. Since European colonization of North America, natural grasslands have declined by over 99% of their historic range, making these one of the most endangered ecosystem types in the Western Hemisphere [6]. Landscape alteration for agricultural practices, excluding naturally occurring, ecologically significant surface fires and clearing land for road and infrastructure construction are the primary causes of prairie loss [1,2,7]. Due to the widespread nature of prairie ecosystems, many of the species comprising a prairie community can be found in a range of latitudes, elevations, and moisture gradients [8,9]. This in turn may influence the evolutionary trajectories of individual species to become genetic variations called ecotypes, that are highly adapted to the conditions of their local environment [9,10]. For restoration project managers working in areas with more extreme settings (i.e., greater altitudes, drier soil conditions, etc.), the question of where to source seeds may be even more crucial [4].

Little bluestem (*Schizachyrium scoparium* (Michaux) Nash) is a warm season perennial bunchgrass in the Poaceae family that is found throughout the continental United States, lower Canadian provinces, and northern Mexico [5,11–14]. Little bluestem is a $C_4$ photosynthetic species, whose highly productive and efficient metabolic rate enables *S. scoparium* to persist in a range of environmental conditions [5,13,14]. Yet, because of its shade intolerance

and smaller growth form, it is generally found in drier, upland, sandy sites, due to increased competition in more mesic areas from larger prairie species [5,7,14]. Due to its wildlife value, erosion control properties, and high net productivity, little bluestem is frequently used in grassland restorations [7,14,15]. However, numerous studies have demonstrated evidence of ecotypic variation within this species, making its use a point of interest in the seed source debate [15–19].

Longleaf pine savannas have been impacted by the same anthropogenic influences as other prairie ecosystems and they currently cover less than 3% of their pre-European range [20–25]. Semi-frequent fires (1–5 years), traditionally ignited by lightning and Native American peoples, are a necessary disturbance for maintaining community health and composition [19,21,23]. These ecologically significant fires are carried by the species rich understory, in particular the bunchgrasses, who move the fires across hundreds of miles of forest floor, creating a mosaic effect in the landscape that contributes to the overall biodiversity of the region [1,15,23,25,26].

For much of the longleaf pine savanna's historic range, the main bunchgrass species with this responsibility was wiregrass (*Aristida stricta* Michx. and *A. beyrichiana* Trin. and Rupr.) [24,27,28]. However, there is a strip through central South Carolina (SC) where wiregrass is not present and little bluestem is the primary bunchgrass species [27–30]. This area, running from the northwestern corner of SC to the coast from 35° N to 33° N latitudes, has been dubbed the "Wiregrass Gap" [22,27,28]. The longleaf pine savanna ecosystem dominates in a large part of the lower elevation regions of the "Wiregrass Gap" [19,22,26]. Due to the extreme scale of habitat loss experienced by this ecosystem, many public and private landowners of degraded pastureland and loblolly plantations have been conducting restoration projects in an effort to reinstate the vegetation communities and fire regime characteristics to this endangered forest type [19,23].

The Camden Battlefield and Longleaf Pine Preserve (Battlefield) is a 476 acre property with stands of a 20 year old loblolly pine plantation (Figure 1). Camden Foundation's main management objective after acquiring the property in 2017 was to restore the property to the ecosystem that was present during the Battle of Camden, a Revolutionary War battle that occurred at the site on the 16th August 1780 [31,32]. Camden, SC, is located in the Carolina Sandhills, an ecologically and edaphically unique ecotype formed by the beaches of Miocenic oceans, that exhibits xeric, sandy soil types, such as Ultisols and Entisols [16,19]. The battlefield is also located within the boundaries of the Wiregrass Gap. A top priority of the restoration is to increase the little bluestem population [31].

In areas where local plant diversity has been severely degraded due to anthropomorphic forces, reseeding efforts are an important consideration to ensure restoration success [4–7,15]. To increase germination of the broadcasted seeds, practitioners utilize different site preparation treatments to decrease competition from undesirable plants and increase exposure to mineral soil [2,6,24]. Two common strategies, both of which increase contact with mineral soil, are raking, which negate above-ground vegetative competition, and discing, which negates both above- and below-ground competition [5,24,28]. Ideally, site preparation measures are selected based on what the environment needs in consideration of the management goals, but oftentimes choices can be limited or even unobtainable due to funding, available equipment, and personnel [1,6–8].

We wished to test whether little bluestem seed source, locally collected or purchased online, would impact establishment, growth, and persistence through the growing season. We further wished to investigate the most appropriate site preparation techniques for restoring little bluestem. Grass seeds are very small and lightweight, and have low quantities of stored nutrients. Many grassland restoration projects recommend disturbing the ground layer to expose mineral soil, which increases the likelihood of germination [5,28]. Some suggestions for soil disturbance and regulating current vegetative competition include discing, raking, or prescribed fire [1,5,33,34]. We conducted a small-scale field experiment comparing the germination and persistence of locally collected and commercially sourced little bluestem seeds over the 2020 growing season in plots that either received discing,

raking, or no site treatment [35]. We also compared root to shoot ratios from local seeds and commercially sourced seeds in a controlled greenhouse experiment. We hypothesize that local seed will have the most and largest plants with the most inflorescence by the end of the growing season. We also hypothesize that plots receiving either of the site preparation treatments will have higher rates of germination and establishment. For the greenhouse experiment, we hypothesize that local seeds will invest more energy in a below-ground root structure compared to the commercially sourced seeds.

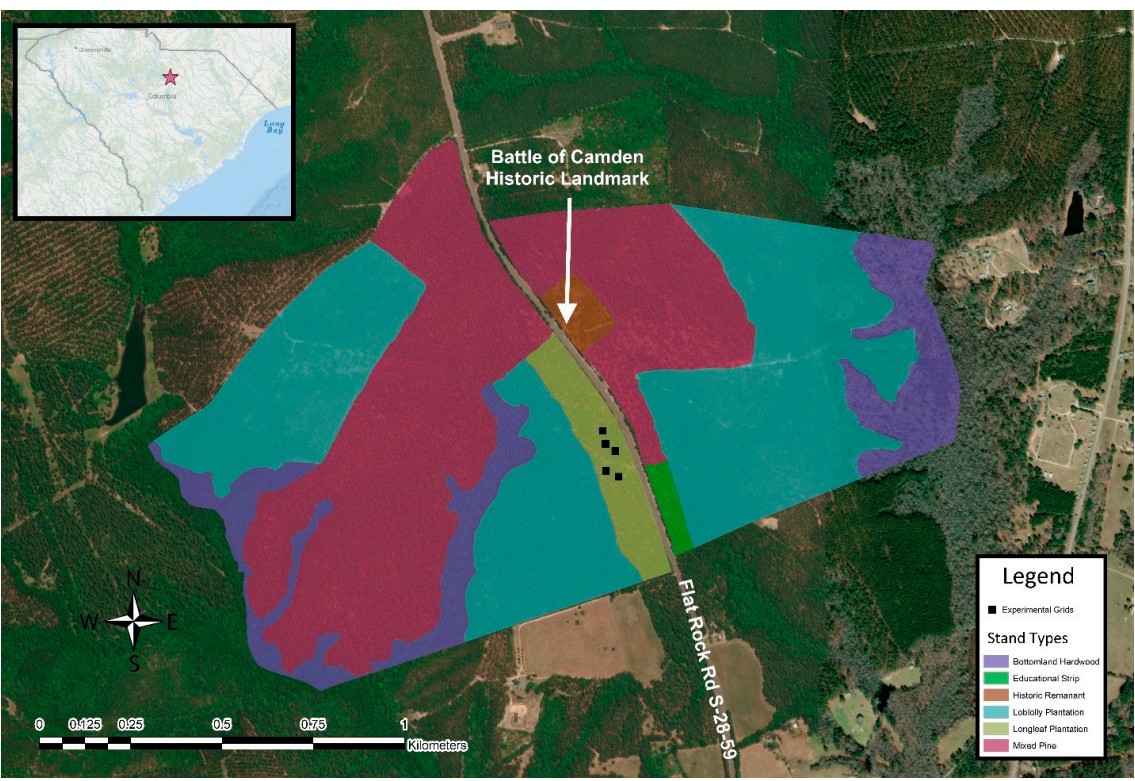

**Figure 1.** Map with aerial imagery displaying the stand types of the Camden Battlefield and Longleaf Pine Preserve, as well as the locations of the five experimental grids in the longleaf pine stand. This area is located north of Camden, SC, which is situated within the ecoregion known as the Carolina Sandhills. (Image courtesy of the author, created through ArcMap 10.8.)

## 2. Materials and Methods

Commercial seeds (RS) were acquired from Roundstone Native Seed (Upton, Kentucky, United States), who produce the seed from populations grown in southern Kentucky (KY). This is the closest large-scale seed producer to the battlefield, and before experimentation, it was the preferred source of the Camden Battlefield restoration project managers [2,31]. Local seed (CAM) was collected from about 30 flowering *S. scoparium* individuals found throughout the study area in November 2019 [36,37]. Seeds were stored unprocessed (i.e., as chaffy seeds, which consist of the caryopsis and its conjoined appendages [37]) in a paper bag indoors (22 °C) throughout the winter until time of sowing.

The field experiment was conducted in a 21 acre stand of plantation-style planted longleaf pine in Camden, SC, United States (age estimated to be 19 years old) (Figure 1). Logging slash, woody debris, and needle accumulation had been mulched the previous year (August 2019), and the understory was undisturbed during the experiment [34,36]. The understory vegetation was sparse and consisted of a mix of grasses and forbs, such as *Eupatorium* spp., *Solidago* spp., *Lespedeza* spp., *Carex* spp., and *Danthonia* spp. Soils at the battlefield are sandy (3.71% organic matter) and acidic (average pH reading 4.578). Camden, SC, experiences a mild climate, the hottest months are July and August with an average daily maximum temperatures of 22 °C, and the coolest months are December and

January, with average daily temperatures of 6.6 °C [38]. The growing season is typically 192 days, starting on April 14 and ending on around October 23 [39]. The average annual precipitation received is 120.269 cm, with April and November being the driest months (both have an average precipitation of 7.5 cm) [38].

Approximately 4 g (4.001–4.0317 g) of CAM and RS seeds was broadcast seeded in plots that received one of three site preparation treatments: digging (Dig), which emulated a discing machine such as a chisel plow or a rototiller; raking (Rake), which emulated a rake attachment for a skid steer; and doing nothing (None), an option often chosen by forest managers with large or steep tracts of land needing reseeding. In total, there were six treatment combinations (CAM-disc, CAM-rake, CAM-none, RS-disc, RS-rake, and RS-none) and these were tested across the stand with five replications of a grid plot design (Figure 2). Grids (2 m × 4 m) were randomly placed and composed of eight 1 m² plots that were randomly assigned one of the treatments or designated as a control (two controls per grid). Animal enclosure boxes were constructed with rough-cut pine boards, aluminum screen cloth (0.635 × 0.635 cm), and polypropylene wildlife netting (1.27 × 1.27 cm). A drip irrigation system composed of 4 L reservoirs (milk jugs) and landscape mesh was installed. A box design aimed to eradicate herbivore and granivore effects while still allowing adequate solar radiation and available moisture to reach the germinating seeds. Water reservoirs were refilled each time plants were measured. Boxes were assembled and partially buried (to prevent burrowing granivores) and the site was prepped a few days before seeds were broadcasted (Figure 3). Shovels and rakes were manually utilized to provide the effects of discing and raking (i.e., eliminate above- and below-ground vegetative competition or eliminate only above-ground competition, respectively).

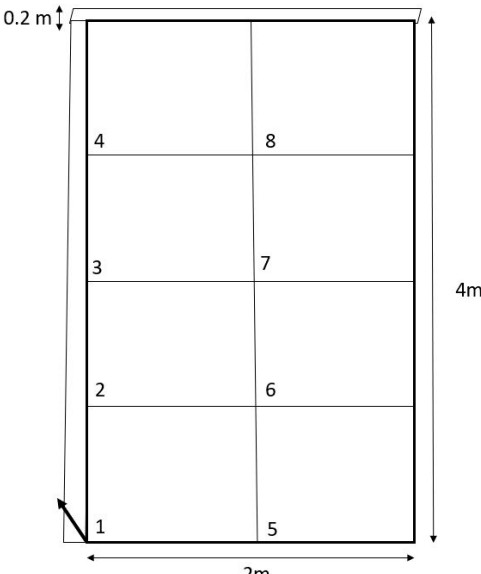

**Figure 2.** Design for the experimental grids. Each box (4 m × 2 m) was composed of eight, 1 m × 1 m treatment plots numbered 1–4 along the left side and 5–8 on the right. Six treatments and two controls were randomly selected across these plots. An enclosure made of raw pine board and aluminum screens encircled the outside of the box, measuring about 0.2 m aboveground and 0.1 m below (to ward off burrowing granivores).

Count and height data were collected for all plots every 2–4 weeks (13–47 days, mean = 23 days) throughout the growing season ($n = 9$). On the final date for data collection, inflorescence counts were recorded and all living plants were harvested, dried, and weighed to compare above-ground biomass across treatments.

Forty 6 in pots were filled with potting soil (SunGro Fafard® 3B Mix Metro-Mix 830) and received 6–10 seeds each [14]. Seedlings were pruned three times (4 weeks, 8 weeks, and 12 weeks) to only have one individual growing per pot. The greenhouse was kept

between 18 and 27 °C with plants being watered every day until 12 weeks, when they received water every other day. At 5 weeks and every 3 weeks thereafter, three random plants from each seed source were harvested and the length of the roots and shoots were measured. Plants were dried for approximately 48 h at 65 °C and then weighed using a fine scale balance (Oahu's Model AS120). Length and mass measurements were utilized to generate root-to-shoot (R-S) ratios to investigate effort invested in above ground structures vs. below ground roots.

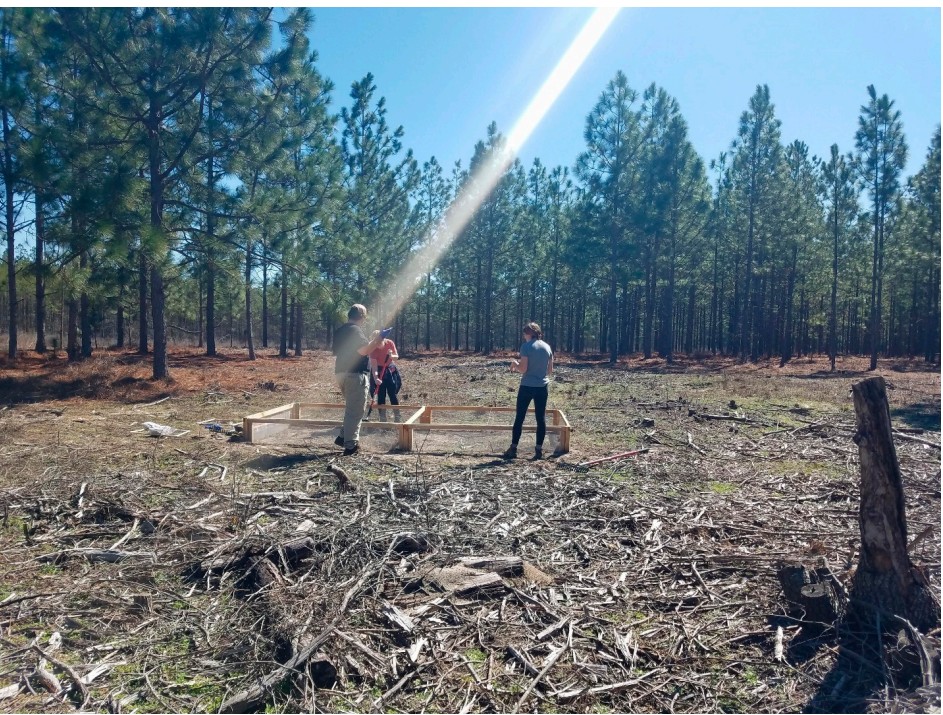

**Figure 3.** An example of an animal enclosure box created for this field experiment. Five boxes composed of eight 1 m by 1 m treatment plots were randomly placed throughout a stand composed of planted 19 year old longleaf pines. Photo courtesy of the authors.

Data collected on count, height, inflorescence, and biomass were analyzed in R (v. 3.6.1), JMP (Pro 14.1.0), and Microsoft Excel (Version 2031). Count data and length data were analyzed with a repeated measures ANOVA, and site preparation and seed source data were analyzed with a two-way ANOVA. Findings were further analyzed with student's t-tests and Tukey–Kramer HSD tests to determine the difference between specific treatment means. Data collected at the final measuring stint (i.e., count, height, above-ground biomass, and inflorescence) were analyzed using an ANOVA. Greenhouse data were analyzed using a one-way ANOVA analysis.

## 3. Results

### 3.1. Question 1: Does Seed Source Matter?

Significant differences in survival between Camden (CAM) and Roundstone (RS) plants (Figure 4) were detected at the end of the growing season ($p$-value $\leq 0.0001$). While there were significantly more RS plants early in the growing season ($p$-value = 0.0056), by the latter half CAM plants persisted and were greater in number ($p$-value $\leq 0.001$, Table 1). RS plants were generally larger, having an average weight of 0.50 g, whereas CAM plants were 0.26 g on average and had an average height of 29.368 cm (RS) versus 20.949 cm (CAM). Inflorescence ($p$-value = 0.86) and biomass ($p$-value = 0.95) differences between the seed sources were insignificant (df = 1). Heights observed were significantly different between seed sources ($p$-value $\leq 0.0001$).

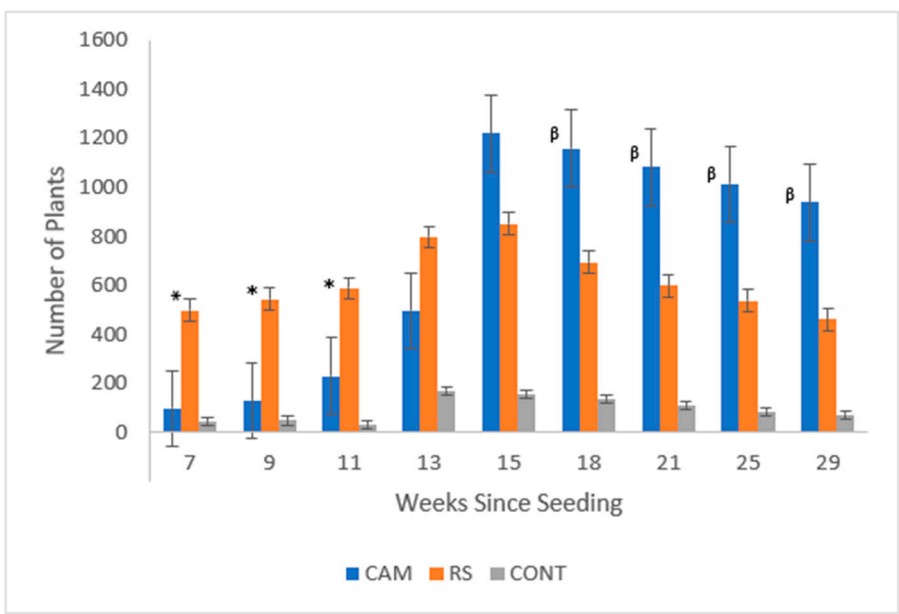

**Figure 4.** Total plants by seed source recorded across the 2020 growing season. Roundstone (RS) initially had significantly more seedlings (denoted by \*) at weeks 7, 9, and 11, but by the end of the growing season, there were significantly more Camden (CAM) seedlings (denoted by $^\beta$) at weeks 18, 21, 25, and 29.

**Table 1.** Plants observed by seed source (source) for the duration of the experiment as analyzed by a repeated measures ANOVA. RS had significantly more plants observed at weeks 7, 9, and 11 (shown with \*), but there were significantly more CAM plants observed at weeks 18, 21, 25, and 29 (shown with $^\beta$). (Degrees of freedom = 2.)

| Weeks | *p*-Value | CAM | | RS | |
|---|---|---|---|---|---|
| | | Average | Standard error | Average | Standard error |
| 7 | 0.0056 * | 7 | 1.871877 | 33 | 9.794654 |
| 9 | 0.0060 * | 10 | 2.95363 | 36 | 10.43949 |
| 11 | 0.0053 * | 15 | 3.692549 | 39 | 10.19393 |
| 13 | 0.2072 | 33 | 7.561725 | 53 | 15.52045 |
| 15 | 0.0699 | 81 | 16.82251 | 57 | 12.73483 |
| 18 | 0.0232 $^\beta$ | 77 | 11.62605 | 46 | 11.34509 |
| 21 | 0.0063 $^\beta$ | 72 | 10.24085 | 40 | 8.2786 |
| 25 | 0.0003 $^\beta$ | 68 | 9.194097 | 36 | 5.679928 |
| 29 | <0.0001 $^\beta$ | 63 | 7.901095 | 31 | 4.932561 |

The data observed at the end of the experiment at 29 weeks post-planting are not reflective of the entire experiment. Throughout the experiment, an accumulative total of 643 RS plants died compared to 477 CAM plants. Height data through the entire growing season between source plots were not significantly different.

One-way ANOVAs with the greenhouse data revealed that RS plants had a significantly greater weight compared to CAM plants in weeks 18 (*p*-value = 0.0325) and 21 (*p*-value = 0.003). RS plants were also significantly longer at week 8 and were almost significantly longer at weeks 11 and 24 (*p*-values = 0.0125, 0.0547, and 0.0549, respectively). Regarding root-to-shoot ratios (R-S), CAM plants had a greater R-S ratio by length for the entire experiment and by weight for the first three measurements (Figures 5 and 6). Weight ratios were determined to be significant at week 11 (*p*-value = 0.0311), whereas length ratios were significant at week 18 (*p*-value = 0.0039). A one-way analysis of the data collected at 5 weeks showed both R-S ratio data to be significant, with CAM plants exhibiting an R-S ratio by length 2.5 times greater than RS plants (respective averages 1.15 vs. 0.46,

*p*-value = 0.0015) and a 3 times greater R-S ratio by weight (respective averages 1.46 vs. 0.45, *p*-value = 0.0358).

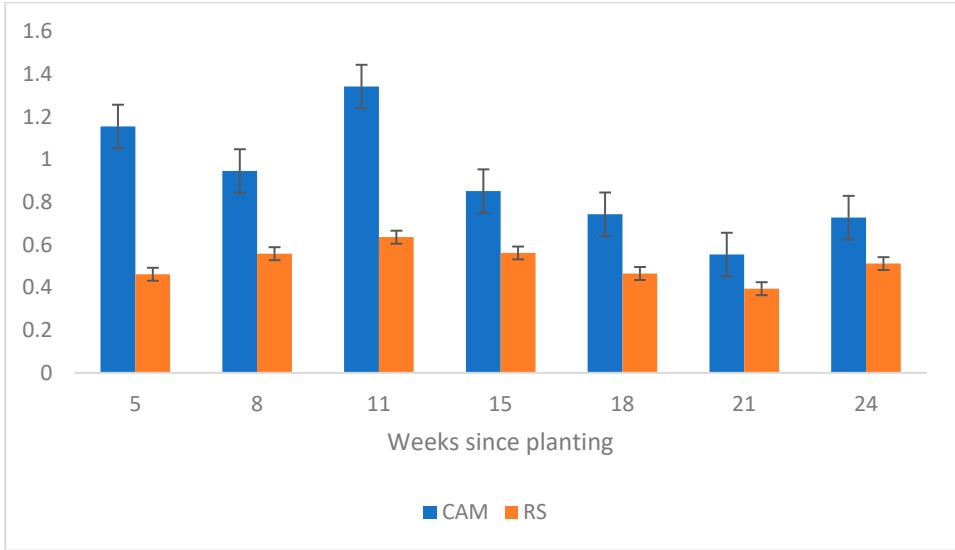

**Figure 5.** Root-to-shoot (R-S) ratios by length determined by the controlled greenhouse experiment for Camden (CAM) plants compared to Roundstone (RS) plants. CAM plants had a significantly greater R-S ratio in terms of length at weeks 5 (*p*-value = 0.0015) and 18 (*p*-value = 0.0039) and CAM plant's R-S ratio was greater in general throughout the entire experiment.

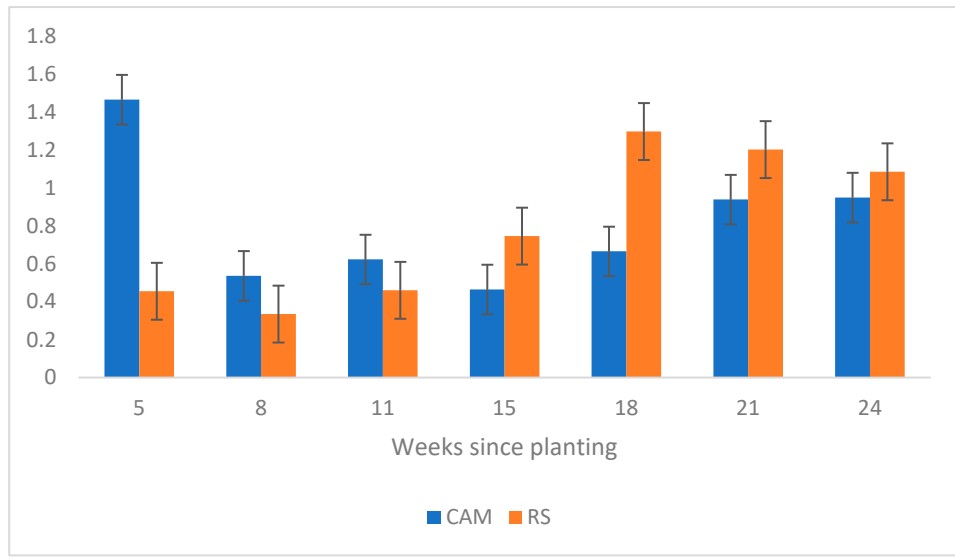

**Figure 6.** Root-to-shoot (R-S) ratios by weight determined by the controlled greenhouse experiment for Camden (CAM) plants compared to Roundstone (RS) plants. Data collected at week 5 showed the R-S ratio by weight exhibited by the CAM plants was 3 times greater than that of the RS plants, and a one-way analysis at week 11 showed CAM plants were significantly different than RS (*p*-value = 0.0311).

*3.2. Question 2: Is Site Preparation Treatment Necessary?*

Data collected at week 29 showed Dig plots had an average of 50 plants, Rake plots had an average of 60 plants, and plots that received no treatment (None) had an average of 48 plants. A Tukey–Kramer HSD test showed all comparisons between treatments had *p*-values greater than 0.82. In terms of inflorescence and biomass data, there was no significant difference for either measurement between treatment plots; significance was only detected when treatment plots were compared to controls.

A repeated measures ANOVA determined treatment plots were significantly different throughout the duration of the experiment (*p*-value = 0.0333). Specifically, significance was detected at weeks 18, 25, and 29 (*p*-values of 0.0413, 0.0124, and 0.0154, respectively). However, this difference was only significant when control plots were factored into the analysis; otherwise, there was no significant difference in the number of plants detected between treatment plots. There was also no significance in the two-way ANOVA conducted to analyze the cumulative effects of seed source and site preparation treatments on the number of seedlings. Rake plots had the greatest number of plants from week 13 onward. Height data between treatment plots were not significant.

## 4. Discussion

The increase in the number of locally sourced CAM plants partway through the growing season indicates the importance of sourcing local seeds. Roundstone seeds initially exhibited a greater germination energy (i.e., the time needed for seedling establishment) than Camden seeds, which is a selling point of commercial seed producers [40]. However, the increase in CAM plants observed midway through the field experiment may be due to CAM plants being better suited for the environmental conditions displayed at the Camden Battlefield. Evidence to support this claim can be seen in the greenhouse data, where CAM plants prioritized root development over growing leaves when initially establishing. This emphasis on root growth over aboveground biomass development may be the pivotal difference between CAM and RS plant survival in dry, low-nutrient environments that occur in the Carolina Sandhills region [19,22,29,33]. This variation between these two ecotype's genetics caused the short-term trend seen in this single growing season and could be the main factor contributing to long-term survival in this environment. Having a well-established root system would allow little bluestem plants to persist regardless of temperature increases and mild drought conditions [5,14]. Additionally, locally collected seeds are chaffy, which helps seeds retain dormancy until environmental conditions are favorable for germination [37].

The locally sourced CAM seeds' success supports claims by other studies that the range of little bluestem is composed of physiological and phenological diverse populations [10,15–19]. Additionally, Dhillion and Friese (1992) determined little bluestem to be a "highly mycorrhizal" plant, signifying that plants from outside sources would lack local fungal symbiotic relationships necessary for survival [41]. Little bluestem is also generally regarded as difficult to germinate in the field due to limitations in available soil moisture [3,33,40]. Therefore, it is referred to as an episodic germinator, relying heavily on local precipitation rates in the area to be seeded. If we would have relinquished our primitive attempts at irrigation in part or completely, we may not have seen any RS plants in our experiment, as it would have been a true test in the Camden Battlefield's local conditions.

Similar to other grassland systems, the longleaf pine ecosystem traverses a range of environmental conditions. Prioritizing local seed sources may ensure long-term restoration success by establishing self-sustaining populations adequately adapted to individual ecosystems [1,4,7,42]. Utilizing locally collected seeds for restoration projects over ordering from commercial seed companies generally requires more time, funding, and labor, but may prove beneficial to the success of the restoration in the long term [34,36,40]. For sites with difficult environmental conditions, such as incredibly xeric or hydrophilic soils, this extra work may be necessary for restoration success.

Seed source is more impactful in re-establishing little bluestem populations than site preparation treatments. This finding has immediate impacts on not only the restoration plan for the Camden Battlefield, but also potentially on similar grassland restoration initiatives throughout the Carolina Sandhills and the world. Longleaf pine restorations generally involve intensive silvicultural treatments (i.e., thinning and clearcutting), which greatly impact the ground by turning over soil layers, uprooting understory plants, and breaking up woody debris [19–21,24–26]. We believe the disturbance caused by overstory management is enough preparation for understory restoration efforts.

Silvicultural treatments provide enough soil disturbance to reduce competing vegetative presence and increase seed contact with mineral soil, negating the need for treatment applications specific to understory re-establishment [43,44]. The stand where we situated our treatment plots was mulched in the understory prior to experiment establishment (August 2019), and this management operative would have definitely impacted seedling establishment by reducing the woody debris and plant litter on the forest floor [36,41,44].

Considering the trends exhibited between the two seed source seedling counts, we should have taken measurements one or two more times before terminating the experiment to truly capture the entire local growing season. The phenology of the Camden Battlefield's local bluestem populations may be different than those sourced from populations grown at Roundstone Native Seed in Hardin County, KY, and larger CAM plants with more seedheads may have been observed if the experiment had been extended. Likewise, Mijnsbrugge et al. (2010) claims that local seeds may take a couple of years to outperform non-local or commercial plants in both size and reproduction [36].

When considering pioneer species, such as the grasses and forbs that generally compose prairie environments, there is still a lot of potential to collect locally from small strips of undisturbed land such as roadsides and fallow pastures. Based on our results, we recommend that land managers emphasize locating and collecting seeds from neighboring, healthy, genetically diverse populations of their focal species rather than putting time and resources into prescribing any site preparation treatments such as discing or raking. We argue that due to the intensive silvicultural treatments required in longleaf pine restorations, the ground will be disturbed enough to have ideal conditions for herbaceous species establishment.

## 5. Conclusions

Our data provide evidence that the seed source should be taken into consideration when reseeding little bluestem in the Carolina Sandhills. In this small-scale ecotypic variation study, we found that during our last measuring stint at the beginning of October, there were significantly more locally sourced plants present than commercially sourced ones. This significance has direct management implications not only for the Camden Battlefield's restoration plan, but for other similar projects occurring in this region and grassland restoration projects in general. The environmental conditions of the Carolina Sandhills are unique compared to the immediately adjacent ecoregions, and special consideration should be given to choosing seeds from populations established under similar conditions. Due to the piecemeal state of longleaf pine ecosystems throughout the southeastern U.S., there are few unaltered vegetative communities to pull from. However, when considering pioneer species, such as the grasses and forbs that generally compose prairie environments, there is still a lot of potential to collect locally in small strips of undisturbed land along roadsides and fallow pastures. Based on our results, we recommend that land managers emphasize locating and collecting seeds from neighboring, healthy, genetically diverse populations of their focal species rather than putting time and resources to prescribing any site preparation treatments such as discing or raking. We argue that due to the intensive silvicultural treatments required for longleaf pine restorations and other such grassland restorations that require greatly reducing the woody overstory density, the ground will be disturbed enough to provide ideal conditions for pioneer herbaceous species establishment.

**Author Contributions:** E.J. and A.H.H. outlined and designed the research purpose and execution; E.J., A.H.H. and P.H. collected materials and constructed experiments; E.J. collected data; E.J. and A.H.H. analyzed data; E.J., A.H.H. and P.H. wrote and edited the manuscript. All authors have read and agreed to the published version of the manuscript.

**Funding:** This research received no external funding.

**Institutional Review Board Statement:** Not applicable.

**Informed Consent Statement:** Not applicable.

**Data Availability Statement:** Data available on request due to restrictions.

**Acknowledgments:** We would like to thank the Historic Camden Foundation, Forest Land Management Inc., Clemson University's Creative Inquiry program, and all of the undergraduate students that helped with this research, including Caleb Scercy, Rosa Kome, Savannah Seeber, Rachel Parnell, Josh Smith, Jackson Smith, Madison LaSala, Corrina McCleod, Emily Jordan, Devin Orr, and Adam Smallridge.

**Conflicts of Interest:** The authors declare no conflict of interest.

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
