# Peer review of "Seed Source for Restoration: Little Bluestem (Schizachyrium scoparium (Michx.) Nash) and the Carolina Sandhills"

_2673-4133, doi:10.3390/ecologies4020022_

Round 1
Reviewer 1 Report
The subject of the paper is very interesting and still poorly recognized. The authors emphasize that natural grasslands are one of the most endangered ecosystem types in the Western Hemisphere. Landscape alteration for agricultural practices and clearing land for road and infrastructure construction are the primary causes of prairie loss.
It seems appropriate to include little bluestem for grassland restorations due to its conservation value, erosion control properties and high net productivity. Evidence of ecotypic variation within this species, making its use a point of interest in the seed source debate. A top priority of the restoration is to increase the little bluestem population.
The studies are original. The paper presents the research on a little bluestem establishment, development and growth of seedlings , and the persistence of plants throughout the growing season depending on the seed source, collected locally or purchased online. Also, the authors determined which site preparation techniques are most appropriate for little bluestem restoration.
The chapters introduction and discussion are well written.
The study was designed correctly. It is a pity that the results are only from one growing season and from one series of pot experiments. Due to changing weather conditions, the results of 3-year field research are usually published. The results are properly interpreted but they need a little improving.
I propose to slightly modify the chapters 2 and chapter 3.
- Figure 1, Figure 2 and Figure 3 should be placed in chapter 2 Materials and Methods. In addition, this chapter should refer to all of these Figures.
- However, in the content of the Results chapter 3, there should be a reference to Table 1.
- The captions of all Figures and Table should not contain a description of the results.
- There is no description and no table or figure with weather conditions (temperature, precipitation) during the field experiment. To be entered as a sub-chapter of chapter 2.
In addition, there is no Conclusion chapter, please include.
Author Response
We thank the reviewer for their time and appreciate their help improving our manuscript. Below we have copied the reviewers comments and explained how we addressed them.
Figure 1, Figure 2 and Figure 3 should be placed in chapter 2 Materials and Methods. In addition, this chapter should refer to all of these Figures.
Done
- However, in the content of the Results chapter 3, there should be a reference to Table 1.
Added
- The captions of all Figures and Table should not contain a description of the results.
Captions have been shortened and interpretations removed.
- There is no description and no table or figure with weather conditions (temperature, precipitation) during the field experiment. To be entered as a sub-chapter of chapter 2.
Weather conditions for the site have be added to Materials and Methods paragraph 2.
In addition, there is no Conclusion chapter, please include.
Done, lines 319-340
Reviewer 2 Report
The introduction does not state what the benefits of site preparation would be and what experience has been gained with this and other species. Because it is not clear why you want to test these treatments.
My main concern is that in materials and method, it is stated that six treatment combinations (2 provenances x 3 site preparation treatments) were performed, but the analysis is done separately. In this case, a two-way ANOVA should be tested to test the effect of provenance, treatments, and their interaction.
Finally, the results and discussion focus on the differences between provenances and leave aside site preparation. More relevant information should be extracted by analyzing the data correctly.
Author Response
We thank the reviewer for their time and appreciate their help improving our manuscript. Below we have copied the reviewers comments and explained how we addressed them.
The introduction does not state what the benefits of site preparation would be and what experience has been gained with this and other species. Because it is not clear why you want to test these treatments.
A discussion of the benefits of site preparation has been added to lines 319-340.
My main concern is that in materials and method, it is stated that six treatment combinations (2 provenances x 3 site preparation treatments) were performed, but the analysis is done separately. In this case, a two-way ANOVA should be tested to test the effect of provenance, treatments, and their interaction.
We did do a two-way ANOVA but the results were not significant so we focused on the provenance analysis which did show significance. We have clarified (methods lines 165-166) and added the results of the two-way ANOVA to the results section, lines 227-229.
Finally, the results and discussion focus on the differences between provenances and leave aside site preparation. More relevant information should be extracted by analyzing the data correctly.
A two-way ANOVA was used and there were no significant differences in site preparation. Therefore, we focused the results and discussion on provenance as significant differences were found there. We have clarified this in the methods and results (see above comment).
Reviewer 3 Report
Row 42: after the full scientific name, indicate that it belongs to graminoid plants, or to the Poaceae family; row 43: "contingent United States" is it typing error instead "continent" or continental"?
Why the subchapter 3.2. Figures, Table and Schemes in the paper? According to Instructions for Authors:
- All Figures, Schemes and Tables should be inserted into the main text close to their first citation and must be numbered following their number of appearance (Figure 1, Scheme I, Figure 2, Scheme II, Table 1, etc.).
Please, position the figures according to instructions in the chapters: Material and Methods, and Results.
Chapter: References needs detailed corrections, because too many journals are referenced under the full name, not with journal abbreviation according to Instructions for Authors: References should be described as follows, depending on the type of work:
- Journal Articles:
1. Author 1, A.B.; Author 2, C.D. Title of the article. Abbreviated Journal Name Year, Volume, page range.
The research presented in paper provides new knowledge on the plant ecology, and efforts to restore parts of the prairies ecosystems in USA. The field investigations and laboratory experiment with seed were appropriately designed, and quality data sets had been analysed and presented to reader. However, the quality of the submitted paper is not good at this stage, and needs substantial corrections.
Author Response
We thank the reviewer for their time and appreciate their help improving our manuscript. Below we have copied the reviewers comments and explained how we addressed them.
Row 42: after the full scientific name, indicate that it belongs to graminoid plants, or to the Poaceae family; row 43: "contingent United States" is it typing error instead "continent" or continental"?
Addressed in line 43 of the introduction
Why the subchapter 3.2. Figures, Table and Schemes in the paper? According to Instructions for Authors:
- All Figures, Schemes and Tables should be inserted into the main text close to their first citation and must be numbered following their number of appearance (Figure 1, Scheme I, Figure 2, Scheme II, Table 1, etc.).
Corrected
Please, position the figures according to instructions in the chapters: Material and Methods, and Results.
Done
Chapter: References needs detailed corrections, because too many journals are referenced under the full name, not with journal abbreviation according to Instructions for Authors: References should be described as follows, depending on the type of work:
- Journal Articles:
1. Author 1, A.B.; Author 2, C.D. Title of the article. Abbreviated Journal Name Year, Volume, page range.
We used the following website to find journal abbreviations (https://images.webofknowledge.com/images/help/WOS/E_abrvjt.html). Not all of the journals we cited could be found on the site. We have fixed those that we could.
Round 2
Reviewer 1 Report
Line 116-118 – “Seeds were stored ……. until time of planting” Wrong word – seeds are sown and seedlings are planted.
Liine 128 – “the coolest months are December and January, with average daily temperatures of 46.5áµ’C and 44.2áµ’C, respectively [37].” I wanted to check these temperatures and tried to open website [37] - but it is impossible "not found". In my opinion such temperatures in December and January are impossible (inoC).
Author Response
We thank the reviewer for catching these errors.
Line 116-118 – “Seeds were stored ……. until time of planting” Wrong word – seeds are sown and seedlings are planted.
We changed planting to sowing on line 116
Liine 128 – “the coolest months are December and January, with average daily temperatures of 46.5áµ’C and 44.2áµ’C, respectively [37].” I wanted to check these temperatures and tried to open website [37] - but it is impossible "not found". In my opinion such temperatures in December and January are impossible (inoC).
Temperatures were reported in Fahrenheit instead of C. We have corrected this and put the temp in C. lines 126-127
Reviewer 2 Report
The corrections and clarifications are sufficient, the manuscript improved substantially.
Reviewer 3 Report
The research presented in paper provides new knowledge on the plant ecology, and efforts to restore parts of the prairies ecosystems in USA. The field investigations and laboratory experiment with seed were appropriately designed, and quality data sets had been analysed The quality of the paper is significantly improved, the authors accepted and applied recommendations given by the Reviewer.
Author Response
The research presented in paper provides new knowledge on the plant ecology, and efforts to restore parts of the prairies ecosystems in USA. The field investigations and laboratory experiment with seed were appropriately designed, and quality data sets had been analysed The quality of the paper is significantly improved, the authors accepted and applied recommendations given by the Reviewer.
We thank the reviewer for their time!